# Development of Hydroxamic Acid Compounds for Inhibition of Metallo-β-Lactamase from *Bacillus anthracis*

**DOI:** 10.3390/ijms23169163

**Published:** 2022-08-15

**Authors:** Andrew E. Huckleby, Jhawn G. Saul, Hyunshun Shin, Staci Desmarais, Apparao Bokka, Junha Jeon, Sung-Kun Kim

**Affiliations:** 1Department of Natural Sciences, Northeastern State University, Broken Arrow, OK 74014, USA; 2Department of Chemistry and Biochemistry, McMurry University, Abilene, TX 79697, USA; 3Department of Chemistry and Biochemistry, The University of Texas, Arlington, TX 76019, USA

**Keywords:** hydroxamate, enzyme kinetics, molecular dynamics, lactamase, inhibition

## Abstract

The emergence of resistant bacteria takes place, endangering the effectiveness of antibiotics. A reason for antibiotic resistance is the presence of lactamases that catalyze the hydrolysis of β-lactam antibiotics. An inhibitor of serine-β-lactamases such as clavulanic acid binds to the active site of the enzymes, thus solving the resistance problem. A pressing issue, however, is that the reaction mechanism of metallo-β-lactamases (MBLs) hydrolyzing β-lactam antibiotics differs from that of serine-β-lactamases due to the existence of zinc ions in the active site of MBLs. Thus, the development of potential inhibitors for MBLs remains urgent. Here, the ability to inhibit MBL from *Bacillus anthracis* (Bla2) was investigated in silico and in vitro using compounds possessing two hydroxamate functional groups such as 3-chloro-N-hydroxy-4-(7-(hydroxyamino)-7-oxoheptyl)benzamide (Compound **4**) and N-hydroxy-4-(7-(hydroxyamino)-7-oxoheptyl)-3-methoxybenzamide (Compound **6**). In silico docking and molecular dynamics simulations revealed that both Compounds **4** and **6** were coordinated with zinc ions in the active site, suggesting that the hydroxamate group attached to the aromatic ring of the compound plays a crucial role in the coordination to the zinc ions. In vitro kinetic analysis demonstrated that the mode of inhibitions for Compounds **4** and **6** were a competitive inhibition with *K*i values of 6.4 ± 1.7 and 4.7 ± 1.4 kcal/mol, respectively. The agreement between in silico and in vitro investigations indicates that compounds containing dihyroxamate moieties may offer a new avenue to overcome antibiotic resistance to bacteria.

## 1. Introduction

Anthrax is an infectious disease caused by the Gram-positive bacteria *Bacillus anthracis*. This infection is commonly found in mammals; however, it could be fatal if the inhalation infection cases are not treated soon after exposure [1,2]. An infection of *B. anthracis* used to be treated with a class of β-lactam antibiotics, but the increased occurrence of antibiotic-resistant strains has made treating such infections increasingly difficult. This emergence of antibiotic resistance has been partially attributed to the production of β-lactamase enzymes capable of hydrolyzing the β-lactam ring within many types of these common antibiotics, and thereby rendering them ineffective in fighting off infection. Today the different variants of β-lactamases discovered can be categorized into the four groups A, B, C, and D [3]. Groups A, C, and D make up a class of β-lactamases called serine-β-lactamases (SBLs) [3]. This class relies on an active site serine residue that functions as the nucleophile in hydrolysis of the β-lactam substrate [3]. The group B β-lactamases, commonly referred to as metallo-β-lactamases (MBLs), require the presence of Zn(II) ions within the active site to carry out catalytic activity [3]. This group of MBLs can be further divided into three subgroups (B1, B2, B3) that vary primarily in substrate preference and the number of Zn(II) ions they require [4]. The B1 subgroup can require one or two Zn(II) ions, referred to as Zn_1_ and Zn_2_. The B2 subgroup generally prefers only one Zn(II) ion, and the B3 subgroup often requires two Zn(II) ions where Zn_2_ is stabilized by an additional His residue in place of a Cys [4]. The Sterne strain of *B. anthracis* produces an MBL of the B1 subtype, referred to as Bla2 [5,6]. The Bla2 active site can contain both the tightly bound Zn_1_ and loosely bound Zn_2_, with the presence or absence of Zn_2_ having no significant impact on catalytic activity [3].

Some clinically available β-lactamase inhibitors, such as clavulanic acid and tazobactam, are used to inhibit a wide array of SBLs but remain ineffective against MBLs today [3]. This is due in part to the broad substrate profile that MBLs show in hydrolyzing β-lactam antibiotics [7]. Given that MBLs rely on the presence of Zn(II) ions within their active sites, a promising approach to developing broad-spectrum MBL inhibitors could be chelating of the active site Zn(II) ions. It has been shown that metalloproteinases can be effectively inhibited by compounds containing a hydroxamic acid group that binds the catalytically crucial zinc ions, and thereby inactivating the enzyme [8]. The use of hydroxamic acid compounds was further investigated in the context of inhibiting *Aeromonas hydrophila*, FEZ-1, and *B. anthracis* MBLs, all with promising results [5,9,10]. Here we seek to further explore the potential for two newly-developed hydroxamic acid compounds to inhibit Bla2 activity by way of binding active site Zn(II) ions. The ability of each compound to effectively bind the Zn(II) ions within the active site was studied by methods of in silico and in vitro analysis. We performed a series of molecular dynamics (MD) simulations of Bla2 in complex with the newly developed hydroxamic acid inhibitors and the common hydroxamic acid compound suberoylanilide hydroxamic acid (SAHA). Here we report the results of three simulations in which enzyme-inhibitor stability and key interactions are compared, and later kinetic analysis that was used to experimentally test the ability of each compound to inhibit Bla2.

## 2. Results

### 2.1. Synthesis of Compound ***4*** and ***6***

We previously synthesized Compound **2** without any substituents as zinc-dependent enzymes including inhibitors for MBLs [5]. Syntheses of Compound **4** with meta-chloro and Compound **6** with meta-methoxy on benzohydramate moiety were similar to its Compound **2** (Figure 1) [5]. Subsequent reaction of either **3** or **5** with aqueous hydroxylamine (3 eq) and potassium hydroxide (4 eq) in the presence of methanol yielded di-hydroxamic acid derivatives **4** and **6** in good yields (72 and 66%, respectively). These compounds were synthesized to learn the influence of enzymatic activities and selectivity at the active site of Bla2 due to their different sizes and polarities.

### 2.2. Molecular Docking

To obtain models of the inhibitors Compound **4**, Compound **6**, and SAHA in complex with Bla2 the molecular docking simulations were carried out. The docking result of lowest energy was selected from each complex and used for further dynamics study (Table 1).

In reviewing the conformation for the complex chosen for 3-chloro-N-hydroxy-4-(7-(hydroxyamino)-7-oxoheptyl)benzamide (Compound **4**) and N-hydroxy-4-(7-(hydroxyamino)-7-oxoheptyl)-3-methoxybenzamide (Compound **6**), a hydroxamate group attached to the aromatic end (referred to as the aromatic hydroxamate group) can be found binding both Zn(II) ions. The hydrophobic nature of the active site residues F63 and V68 seem to form a more nonpolar environment favoring the aromatic end of the compounds. In addition, several potential hydrogen bonds were identified between the compounds (Compound **4** and Compound **6**) and the active site residues, with N209 forming a potential hydrogen bond with the aromatic hydroxamate group, and K200, S201, and H239, forming a potential hydrogen bond with a hydroxamate group attached to the aliphatic end, referred to as the aliphatic hydroxamate group (Figure 1A,B). The side chain of residue N209 is also within range to form a hydrogen bond with the methoxy of Compound **6** (Figure 1B). For the SAHA conformation, the aliphatic hydroxamate group is bound to the Zn(II) ions and also could potentially form hydrogen bonds with N209. The amide group toward the aromatic end of SAHA also forms a potential hydrogen bond with K200. In reviewing all of the intermolecular interactions, most of the stability of SAHA within the Bla2 active site comes from the hydrophobic interactions between the aliphatic backbone of SAHA and the nonpolar active site residues F63, W88, and V68 (Figure 1C).

### 2.3. Molecular Dynamics

A comparison of the Bla2 backbone RMSDs between the simulations demonstrates that Bla2 stability is achieved around 0.17 nm in relation to the energy minimized conformation, indicating that the systems are sufficiently equilibrated within at most 5 ns (Figure 2A). The Bla2 backbone RMSDs for all three systems were relatively similar (Figure 2A). In comparing the RMSDs of the Compounds themselves (Figure 2B), one can see that the Bla2:Inhibitor complex for each system is maintained throughout the simulation. It should be mentioned here that the RMSD of SAHA appears to be rather different from the cases of Compounds **4** and **6**. The possible explanation would be that the aliphatic hydroxamate of SAHA binds to the zinc ion whereas the binding hydroxamate of Compounds **4** and **6** is opposite. In addition, we performed a 70 ns simulation of Bla2 alone to further investigate the stability of the protein. Given the radius of gyration of the Bla2 backbone structure in the 70 ns simulation, the structure remains stable throughout the simulation process, indicating no significant unfolding of the protein structure; furthermore, the RMSD of the Bla2 structure in the 70 ns simulation revealed that the stability of the protein backbone is maintained around 0.175 nm (Appendix A). Both Compounds **4** and **6** deviate only within a range of 0.15–0.3 nm in relation to the protein active site.

In comparing this analysis to the RMSD of SAHA, minimal difference in complex stability is seen. To develop a more detailed understanding of the interactions taking place between Bla2 and the inhibitors, the decomposition data (Appendix A) for each system were analyzed. In reviewing the dynamics for Compound **4**, the aromatic hydroxamate group remains bound to both Zn(II) ions. Throughout the simulation, a potential parallel π-stacking interaction between the aromatic ring of Compound **4** and the side chain of H239 is sustained (Figure 3A).

This π-stacking interaction and the nonpolar environment provided by active site residues, are the likely reason for the stability and lack of torsion throughout the simulation for the aromatic end of Compound **4**. The aliphatic hydroxamic group continued to form frequent hydrogen bonds with the side chain of K200, backbone of H239, and the side chain of S201 (Figure 3A). The most dominant residues in stabilizing Compound **4** were H239 with the π-stacking interactions and hydrogen bonding, in addition to the hydrogen bonding of the K200 side chain (Appendix A). The same π-stacking interaction of H239, and hydrophobic surrounding of F63 and V68, can be found stabilizing the aromatic ring of Compound **6** (Figure 3B). Similar to Compound **4**, the aromatic hydroxamate group of Compound **6** remained bound to both Zn(II) ions throughout the simulation. The aliphatic hydroxamic group of Compound **6** manages to sustain a noticeably stable network of hydrogens bonds between the K200, S201, and H239 (Figure 3B). In reviewing the dynamics for SAHA, the aliphatic hydroxamic group does remain bound to both Zn(II) ions. The aromatic end of SAHA associates mostly with peripheral residues of the Bla2 active site (Appendix A). The stability of SAHA within the Bla2 active site seems to be sustained primarily by way of hydrophobic interactions with the F63 and V68 residues. Although it does form occasional hydrogen bonds with H239 and K200, these interactions with Bla2 do not appear to be as stable as they are for Compounds **4** and **6**. In calculating the end-state free energy of each simulation, the free energy change of the Compound **6**:Bla2 complex was the most stable (Table 1), while the free energy changes calculated for the Compound **4**:Bla2 and SAHA:Bla2 complexes were higher in value and relatively similar to one another (Table 1). Looking further into the free energy components for each system, Compounds **4** and **6** had electrostatic energy contributions of −32.50 ± 1.05 kcal/mol and −38.96 ± 1.18 kcal/mol, respectively. This is in comparison to the electrostatic energy contribution of SAHA at −0.16 ± 0.98 kcal/mol.

### 2.4. Inhibition Tests

To explore the possibility of inhibition of Bla2 activity by Compound **4** and **6**, IC_50_ values were determined. As a control experiment, SAHA was used due to the presence of a hydroxamate functional group in the compound. Concentrations of Compound **4** and **6** were used from a range of 0.5 μM to 100 mM to gain IC_50_ values. All the data obtained were fit to a concentration-response plot with the equation *v*_i_/*v*_o_ = 1/(1 + ([I]/IC_50_)^h^), where I is an inhibitor and *h* is the Hill coefficient (the Hill coefficients used were between 0.5 and 1). As shown in Figure 4, the data points for compound **4** were well fit to the semilog concentration-response plot, and the IC_50_ was able to be calculated from the plot at 50% inhibition: that is, 20.0 ± 5.0 μM. Compound **6** behaves similarly to Compound **4** with the IC_50_ value of 14.9 ± 9.8 μM (Figure 4).

However, the IC_50_ value for SAHA was higher than 100 μM, which is too high to be considered as a drug candidate (Appendix A).

Attempts were made to explore the mode of inhibition, inhibitory enzyme assays were carried out using various concentrations of the substrate nitrocefin as well as Compound **4** and **6**. Figure 5 shows Lineweaver–Burk plots with a nest of lines that intersect at the *y*-axis for both Compound **4** and **6**. *K*_i_ values were determined on the basis of the plots with a *K*_i_ value of 6.4 ± 1.7 μM for Compound **4** and a *K*_i_ value of 4.7 ± 1.4 μM for compound **6**, where *K*_i_ values were calculated from the slope based on *K*_m_ and apparent *K*_m_ values were obtained from Figure 5. The IC_50_ and *K*_i_ values for Compound **4** and **6** and SAHA are listed in Table 2.

## 3. Discussion

We have demonstrated the potential for inhibition of Bla2 by two hydroxamate compounds that have shown the ability to bind both of the active site zinc ions. The effectiveness of the zinc ion binding was investigated by way of in silico and in vitro analyses. The MD simulations of Bla2 in complex with the newly developed hydroxamate compounds and the control compound SAHA further revealed the details of the binding interactions. In addition, kinetic analyses were performed to examine the inhibitory potential of each compound. The performance of the MD simulations for both the Bla2:Compound **4** complex and Bla2:Compound **6** complex, clearly showed the ability of the aromatic hydroxamate group to remain bound to the two active site zinc ions. The binding affinities obtained from Autodock Vina for both Compounds **4** and **6** are comparable at −7.5 and −7.7 kcal/mol, respectively. The free energy values obtained from the MD simulations would show a similar trend with only a slight improvement for Compound **6** as compared to Compound **4** and SAHA. This difference could be attributed to the structures of each compound, concerning the meta-oriented chlorine of Compound **4**, and methoxy of Compound 6. In analyzing the MD simulations, and per-residue decomposition data of each compound, the methoxy of Compound 6 does seem to provide it an exclusive advantage in interacting with the V68 side chain in a variety of ways. This is an interaction that the chlorine of Compound **4** does not allow to occur too frequently. In comparing the results of Compounds **4** and **6** to SAHA, the lack of an electrostatic contribution in the SAHA simulation is certainly more characteristic of its hydrophobic structure and behavior found in the MD simulation, as compared to the more polar structures of Compounds **4** and **6**. This can be seen in the per-residue decomposition data (Appendix A) as polar interactions are not as dominant for SAHA, as they were for Compounds **4** and **6**. The hydrophobic interactions surrounding the aromatic end of Compound **6** also played an important role in stabilizing the compound within the Bla2 active site, as reflected in the per-residue decomposition data. The in silico data obtained for the SAHA simulation showing inhibitory potential comparable to Compounds **4** and **6** clearly is not reflective of the results obtained from in vitro analysis of SAHA. This is likely due to the active site focused grid box used to carry out the docking protocol. Had the grid box been set up to encompass the entire Bla2 structure, SAHA could have potentially shown a preference for a binding conformation beyond the active site. Thus, the current docking protocol may have forced SAHA into a conformation within the Bla2 active site.

The inhibition studies by experimental and in silico analyses confirmed that the mode of inhibition is highly likely to be competitive. As the MD simulations showed that these compounds strongly bind the active site zinc ions, reaffirming their potential as competitive inhibitors. The IC_50_ values for the compounds range from 15 to 20 μM. Given the overlap in calculated error, these values can be considered comparable. In addition to the IC_50_ values, the results of the measured binding interaction strength (*K_i_* values) only range from 4.7 to 6.4 (kcal/mol). Previously, structurally similar compounds containing hydroxamate functional groups were investigated in our laboratory, where 3-(heptyloxy)-N-hydroxybenzamide displayed no inhibition but N-hydroxy-3-((6-(hydroxyamino)-6-oxohexyl)oxy)benzamide showed significant inhibition [5]. It should be noted that 3-(heptyloxy)-N-hydroxybenzamide has only one hydroxamate located at the distal end of the aliphatic group and N-hydroxy-3-((6-(hydroxyamino)-6-oxohexyl)oxy) contains two hydroxamate groups similar to Compounds **4** and **6**. No or little inhibition by 3-(heptyloxy)-N-hydroxybenzamide and SAHA supports the notion that the hydroxamate attached to the aromatic group in the compounds plays a pivotal role in the inhibition and the binding interactions. Although some IC_50_ values in the low micromolar to nanomolar range were found in mercaptoacetic acid thiol ester compounds for various MBLs and a disubstitued succinic acid compound with various hydrophobic substituents for IMP-1 from *P. aeruginosa* [11,12,13]. The dihydroxamate-containing compounds (Compounds **4** and **6**) should be added to the repertoire of promising inhibitory compounds for MBLs.

In all consideration, Compounds **4** and **6** may be useful drug candidates as well as lead compounds for further development. To further investigate the development of better drug candidates, some structural modifications would be needed with structure-activity relationship analysis. It should also be mentioned here that these compounds should be examined further for pharmacokinetics, cytotoxicity and specificity prior to clinical use.

## 4. Materials and Methods

### 4.1. General Procedures

All chemicals were obtained through Sigma or other quality manufacturers. The enzyme Bla2 was purified as previously described [4,14]. The purity of the product through Ni^2+^ chromatography was determined by 10% SDS-PAGE. If necessary, gel-filtration chromatography was performed for further purification. Higher than 95% pure enzyme was stored at −20 °C in 50 mM 3-(N-morpholino)propanesulfonic acid (MOPS) (pH 7.0), 0.050 mM ZnSO_4_, and 30% (*v*/*v*) glycerol for further experiments. 3-chloro-N-hydroxy-4-(7-(hydroxyamino)-7-oxoheptyl)benzamide (Compound **4**) and N-hydroxy-4-(7-(hydroxyamino)-7-oxoheptyl)-3-methoxybenzamide (Compound **6**) were prepared by dissolving them in dimethyl sulfoxide (DMSO), which was then diluted to various concentrations to assay enzyme activities. The hydroxamate compounds were prepared as previously reported [5].

### 4.2. Molecular Docking Simulations

Given that there is no determined crystal structure of Bla2, homology-modelling was used with the SWISS-MODEL program to build a structure file of the enzyme [15]. The crystal structure of metallo-beta-lactamase BcII (4C09) was used as the template for modelling Bla2, with a final GMQE score of 0.87, and QMEAN score of 1.82 [16]. The structure files of both Compounds **4** and **6**, and SAHA, were generated using the Avogadro program with the General Amber Force Fields for energy minimization [17]. A molecular docking simulation was then performed for each ligand using the program Autodock Vina with complete ligand flexibility and a rigid receptor [18]. A grid box with parameters of (60 Å × 52 Å × 52 Å) was centered around the active site of Bla2 with the *x-y-z* coordinates (12.134, 6.796, 24.713). The docking results of lowest energy for each ligand were then prepared for further analysis through molecular dynamics. The Protein-Ligand Interaction Profiler (PLIP) online server was used to observe the interactions from the Autodock Results, and the ending molecular dynamics conformations [19].

### 4.3. Molecular Dynamics Simulations

All molecular dynamics simulations were carried out using the Gromacs program version 2021.4. The force field used for these simulations was the AMBER99SB-ILDN force field modified with recently published parameters for zinc(II)-binding residues [20,21]. These modifications are important for accurate coordination of the active site zinc ions, helping to improve the stability of the metalloprotein during the dynamics simulation. The water model used was the SPC/E water model. To generate the ligand topologies for Gromacs, the program Acpype was used [22]. The first stage of each dynamics simulation was solvating the system, and then neutralizing it with the addition of three chlorine ions. Next, an energy minimization was carried out for 50,000 steps using the steepest descent algorithm. The first phase of equilibration was carried out in a 100 ps run using the *NVT* ensemble with the leap-frog integrator. This run stabilized the temperature around 300 K. The second phase of equilibration was carried out over a 100 ps run using the *NPT* ensemble with the leap-frog integrator as well. This run stabilized the pressure of the system around 1 bar. Position restraints were applied on both the receptor and the ligand for each equilibration run. The temperature was regulated using the Berendsen V-rescale thermostat, and the pressure was regulated using the Parrinello-Rahman barostat. The final production MD simulation was run for 20 ns, during which the position restraints were removed. Two replicate simulations were carried out for each Bla2:Inhibitor complex system with different initial velocities. The replicates were then compared to help identify behavior consistent of each system, and from which the most fit replicate was used for further analysis and comparison of inhibitors. The root mean square deviation (RMSD) of each inhibitor trajectory in relation to the protein backbone was calculated using the RMSD module of Gromacs. In addition, a calculation of the protein backbone RMSD relative to the energy minimized conformation was performed for each system, to provide insight into the protein stability throughout the simulations (Figure 2).

### 4.4. Free Energy Calculations

To calculate the average end-state free energy of each simulation, the molecular mechanics generalized Born and surface area model (MM-GBSA) was used with the program gmx_MMPBSA [23]. For this calculation, 100 snapshots from each simulation were used. The binding free energy calculation used for this simulation can be summarized as:(1)$$\Delta G_{\mathrm{binding}}=\Delta G_{\mathrm{complex}}-\left(\Delta G_{\mathrm{receptor}}+\Delta G_{\mathrm{ligand}}\right)$$

The $\Delta G_{\mathrm{complex}}$ is the free energy change of the Bla2-Inhibitor complex, $\Delta G_{\mathrm{receptor}}$ is the free energy change of Bla2 alone, and $\Delta G_{\mathrm{ligand}}$ is the free energy change of the ligand alone. The variables composing this calculation for the MM-GBSA model can be further understood as so:(2)$$\Delta G_{\mathrm{binding}}=\Delta E_{\mathrm{gas}}+\Delta G_{\mathrm{solv}}$$

The $\Delta E_{\mathrm{gas}}$ component is the vacuum interaction energy, derived from the summation of the bonded and non-bonded interactions. The bonded interactions are composed of the bond, angle, and dihedral potentials. The non-bonded interactions are composed of the van der Waals and electrostatic contributions.
(3)$$\Delta E_{\mathrm{gas}}=\Delta E_{\mathrm{bonded}}+\Delta E_{\mathrm{non}-\mathrm{bonded}}=\left[\Delta E_{\mathrm{bond}}+\Delta E_{\mathrm{angle}}+\Delta E_{\mathrm{dihed}}\right]+\left[\Delta E_{\mathrm{vdw}}+\Delta E_{\mathrm{ele}}\right]$$

The $\Delta G_{\mathrm{solv}}$ component is the solvated free energy made up of the polar $\left(\Delta E_{\mathrm{gb}}\right)$ and nonpolar $\left(\Delta E_{\mathrm{surf}}\right)$ solvation components.
(4)$$\Delta G_{\mathrm{solv}}=\Delta E_{\mathrm{gb}}+\Delta E_{\mathrm{surf}}$$

In addition to the free energy calculation, gmx_MMPBSA was also used to carry out a per-residue decomposition of the free energy, to help analyze the energy contributions of key active site residues within 4 (Å) of the ligand.

### 4.5. Inhibition Tests

The purified Bla2 was assayed for substrate activity by adding enzyme at a final concentration of 0.13 μg/mL to solutions ranging from 10 to 70 μM nitrocefin in 50 mM MOPS (pH 7.0) to a final volume of 1 mL in quartz cuvettes at room temperature. Reactions were monitored by observing the increase in absorbance at 485 nm due to the hydrolysis of nitrocefin (ε = 15,900 M^−1^·cm^−1^) and were performed in triplicate [24]. The assay for IC_50_, the concentration of inhibitor necessary to inhibit 50% of the enzymatic activity, was performed under the aforementioned enzyme assay conditions after incubating the reaction mixture for 1 min with varying concentrations of Compounds **4** and **6** before initiating the reaction with a fixed concentration of nitrocefin (0.4 mM). The enzyme concentration used was 0.13 μg/mL (4.8 nM). A separate series of assays for each inhibitor was run at fixed inhibitor concentrations (0–0.030 mM for compounds **9** and **10**) in which, at each inhibitor concentration, the concentration of the substrate nitrocefin was varied from 0.1 to 0.4 mM. The resulting data were analyzed for mode of inhibition by Lineweaver–Burk plots and were also fit using non-linear regression through SigmaPlot version 11.0 (Sigma, St. Louis, MO, USA), using the competitive inhibition equations: ν = V_max_·S/[K_m_·(1 + 1/K_i_) + S]. All experiments were carried out in triplicate.

## 5. Conclusions

We reported the development of novel di-hydroxamate-containing compounds with high affinity for Bla2, which is a major factor of β-lactam antibiotic resistance. The compounds showed great potential by inhibiting Bla2 with IC_50_ values in the micromolar range and by binding strength with *K*i values in the low micromolar range. MD simulations detail some specific amino acids are strongly engaged in the complex formation of the compounds with the enzyme. In addition, the MD simulations provide insight into the subtle differences that can be found in the interactions between the inhibitors and Bla2. The experimental inhibitory studies support that the mode of inhibition is competitive, which is well consistent with the data obtained from in silico analysis. It is strongly suggested that the hydroxamate group attached to the aromatic ring of the compounds is crucial in the interaction with the zinc ions, enhancing the effectiveness of inhibition. It is expected that the compounds protect β-lactam antibiotics from damage by MBLs.

## Data Availability

Not applicable.

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
