# Peer review of "Development of Hydroxamic Acid Compounds for Inhibition of Metallo-β-Lactamase from Bacillus anthracis"

_ijms, 2022, doi:10.3390/ijms23169163_

Round 1
Reviewer 1 Report
In this manuscript the authors studied two potential inhibitors for β-lactamases from Bacillus anthacis both computational and experimental methods. The work is relevant and can be published but needs, in my opinion, some main revisions to validate the authors' main conclusions. The description of the methodology also needs some minor improvements.
Main revision:
1) Considering that the authors are using a homology modelling structure followed by a docking protocol, my main concern about this work is the simulation time used. I expected large simulation times (min 200ns-500ns) before any conclusion about the interactions between the docked ligands and the studied protein. I also expected some replicas of the MD simulations, with different initial velocities, to prove that all the conclusions are independent from the initial velocities uses. With only 20ns of simulation (1 replica), I would not consider this work to be published. All the analysis should be repeated with longer simulations.
Minor revision:
2) In Figure 2 (and Supplementary Figure 1) the RMSD are plotted for 10ns of simulation, however in the “Materials and Methods” section it is written that the MD simulation production was run for 20ns. The RMSD and RMSF plots should be represented for the complete simulation time.
3) In Figure 3, instead of the ending snapshot of the MD simulation, I would show a representative structure of the all simulation.
4) The structural analysis done on “2.3 Molecular dynamics” section can be complemented with a plot “bond distance vs time” for all the interactions mentioned. The plot will help to understand how the interactions mentioned in the main text change during the MD simulation.
5) Considering that there is no determined crystal structure for Bla2 protein and a homology modelling was created, what is the sequence identity between Bla2 and the metallo-beta-lactamase BcII used as template. Did you considered to use Alphafold2 to predict your structure?
6) Can you provide the homology modeling structure (pdb file) in the supporting information?
7) Can you provide the initial structures used for the MD simulations?
Reviewer 2 Report
In this work, Huckleby and colleagues suggest two new molecules carrying an inhibitor profile against metallo-beta-lactamases, which is important in the bacterial resistance scenario.
The manuscript is well-writen, easy to follow, and the computational protocol, both for molecular docking as for molecular dynamics simulations, is well-conducted.
I will only suggest few points that could improve the overall understanding.
- Table 1 : typo "confirmation" -> conformation.
- Compounds name: please indicate the compounds' names at the beggining of Results' section, in the first time they are mentioned.
- Figure 3: I would rather see the SAHA RMSD together in b) plot for direct comparison, not in the SI.
- Also regarding RMSD, I suggest to present the active centre residues' RMSD, as a form to follow localized conformational changes when felting the presence of the inhibitors.
As you must know AutoDock Vina is not the best docking method to study metallo-enzymes, as will not consider the charge and geometric/coordination features of Zn2+. Of course, one can add the 2+ charge manually before docking run, but I would like to see an explanation to not touse AutoDockZn or other method. Highlight that Zn contribution is taken into account when you choose the AMBER99SB-ILDN force field modified with parameters for zinc(II)-binding residues. This can be done both in methods section as in Discussion.
Would be also interesting to see a pharmacokinetics profile for the two novel inhibitors, easily calculated using an ADMET server, such as ADMETLab or SwissADMET.
Round 2
Reviewer 1 Report
The authors addressed almost all the suggested revisions, which is really appreciated, however I still have some comments/questions before accepting the manuscript for publication.
In my opinion, even if the backbone RMSD seems to be stable after 5ns for two replicas, I would expect a longer MD simulation because the authors are working with a structure from a homology modelling protocol, following by docking. I would like to see at least 100ns, as I have mentioned in the previous report. The authors added 1 replica with 20ns, which have improved the sampling, but it is not enough in my opinion.
Minor revisions:
1. The Figure 2 (and Supplementary Figure 2) x axes were corrected to the entire MD production time (20ns) and the data was represented as an average of two MD replicas, however I am wondering why the B plot is so different when compared to the previous manuscript version. Is the position of Compound 4, 6 and SAHA very different between MD replicas?
2. In the Results sections, 2.3 Molecular Dynamics subsection, the authors have the following sentence: “The same π-stacking interaction of H239, and hydrophobic surrounding of F63 and V68, can be found stabilizing the aromatic ring of Compound 6 (Figure 3B).” However, in the new Figure 3B, the mentioned residues F63 and V68 are not represented. Can you represent these residues as in the previous version (Both for figure A and B)?
3. In the Discussion section the authors mentioned: “In analyzing the MD simulations, and per residue decomposition data of each compound, the methoxy of Compound 6 does seem to provide it an exclusive advantage in interacting with the V65 side chain in a variety of ways.”. Do the authors mean V68 instead of V65, or V65 is another residue not mentioned before?
Author Response
The authors addressed almost all the suggested revisions, which is really appreciated, however I still have some comments/questions before accepting the manuscript for publication.
In my opinion, even if the backbone RMSD seems to be stable after 5ns for two replicas, I would expect a longer MD simulation because the authors are working with a structure from a homology modelling protocol, following by docking. I would like to see at least 100ns, as I have mentioned in the previous report. The authors added 1 replica with 20ns, which have improved the sampling, but it is not enough in my opinion.
Thank you for your suggestions. In order to confirm the stability of the protein over a longer period of time, we performed a 70 ns simulation. We found that there were no significant changes and mentioned that in the text.
Minor revisions:
- The Figure 2 (and Supplementary Figure 2) x axes were corrected to the entire MD production time (20ns) and the data was represented as an average of two MD replicas, however I am wondering why the B plot is so different when compared to the previous manuscript version. Is the position of Compound 4, 6 and SAHA very different between MD replicas?
The difference was explained in the text.
- In the Resultssections, 2.3 Molecular Dynamicssubsection, the authors have the following sentence: “The same π-stacking interaction of H239, and hydrophobic surrounding of F63 and V68, can be found stabilizing the aromatic ring of Compound 6 (Figure 3B).” However, in the new Figure 3B, the mentioned residues F63 and V68 are not represented. Can you represent these residues as in the previous version (Both for figure A and B)?
As suggested, Figure 3B is replaced with a new version with F63 and V68.
- In the Discussionsection the authors mentioned: “In analyzing the MD simulations, and per residue decomposition data of each compound, the methoxy of Compound 6 does seem to provide it an exclusive advantage in interacting with the V65 side chain in a variety of ways.”. Do the authors mean V68 instead of V65, or V65 is another residue not mentioned before?
Thank you for finding the typographical error. It has been fixed.